# Effects of Treatment of Sleep Disordered Breathing on Sleep Macro- and Micro-Architecture in Children with Down Syndrome

**DOI:** 10.3390/children9070984

**Published:** 2022-06-30

**Authors:** Viecky M. P. Betavani, Margot J. Davey, Gillian M. Nixon, Lisa M. Walter, Rosemary S. C. Horne

**Affiliations:** 1Department of Paediatrics, Monash University, Melbourne, VIC 3168, Australia; vieckymirsa29@gmail.com (V.M.P.B.); margot.davey@monashhealth.org (M.J.D.); gillian.nixon@monashhealth.org (G.M.N.); lisa.walter@monash.edu (L.M.W.); 2Melbourne Children’s Sleep Centre, Monash Children’s Hospital, Melbourne, VIC 3168, Australia; 3Department of Paediatrics, Level 5, Monash Children’s Hospital, 246 Clayton Road, Clayton, Melbourne, VIC 3168, Australia

**Keywords:** obstructive sleep apnea, behavior, quality of life, daytime functioning

## Abstract

Background: Children with Down syndrome (DS) are at increased risk of obstructive sleep disordered breathing (SDB), which is associated with intermittent hypoxia and sleep disruption affecting daytime functioning. We aimed to examine the effects of treatment of SDB on sleep quality and daytime functioning in children with DS. Methods: Children with DS and SDB (*n* = 24) completed a baseline and follow-up overnight polysomnographic (PSG) study 22 ± 7 months (mean ± SD) later. Sleep micro-architecture was assessed using EEG spectral analysis, and parents completed a number of questionnaires assessing sleep, behavior, daytime functioning, and quality of life (QOL). Results: A total of nine children (38%) were treated. At baseline, the treated group had more severe SDB compared to the untreated group. SDB severity was significantly improved from 40.3 ± 46.9 events/h to 17.9 ± 26.9 events/h (*p* < 0.01) at follow up in children who were treated. There were no significant differences in sleep macro-architecture parameters from baseline to follow up in either the treated or untreated group. Sleep micro-architecture was not different between studies in the treated group, however this tended to improve in the untreated group, particularly in REM sleep. Daytime functioning and behavior were not different between the studies in either group, however, QOL improved after treatment. Conclusions: Our study identified that treatment of SDB improves severity of the disease as defined by PSG, and this was associated with parental reports of improved QOL, despite treatment having no demonstrable impacts on sleep quality, behavior, or daytime functioning.

## 1. Introduction

The incidence of obstructive sleep apnea (OSA) is far higher in children with Down syndrome (DS), affecting 31–97% of children depending on patient selection criteria, definitions and methodologies used [1], compared to a prevalence in typically developing children of 1–6% [2]. The distinct dysmorphic features of DS, such as mid-face and mandibular hypoplasia, relatively large and medially positioned tonsils, and macroglossia result in a significant reduction in the size of the upper airway in children with DS when compared to typically developing children, thus increasing the risk of OSA [3,4]. Additionally, obesity and hypotonia are common in DS and potentially contribute to the collapse of the upper airway during sleep and the risk of OSA [4]. OSA is at the severe end of a spectrum of respiratory disorders and is characterized by repetitive hypoxia, hypercarbia, and/or sleep disruption. At the mild end of the spectrum, primary snoring (PS) is not associated with significant desaturation or sleep fragmentation [2]. Studies have identified that SDB of all severities, including PS, are associated with adverse effects on daytime behavior and functioning, including poorer school performance in typically developing children [5]. Both parent-reported symptoms [6,7] and PSG studies [8,9,10,11,12] link SDB with reduced daytime executive and language functioning, and cognition in children with DS. It is hypothesized that these adverse outcomes are mediated by the repetitive hypoxia and sleep disruption that are associated with SDB [13]. However, studies using both conventional polysomnographic (PSG) measurements of sleep macro-architecture (the structure of sleep), including total sleep time and the percentage of total sleep spent in the two sleep states (non rapid eye movement (NREM) and rapid eye movement (REM) sleep) have not identified major changes in children with SDB compared to non-snoring control children [14,15]. Power spectral analysis of the EEG obtained during an overnight PSG study quantifies the delta, theta, alpha, and beta waveforms that occur during sleep in the frequency domain (EEG spectral power), and provides a measurement of sleep micro-architecture, which is a more sensitive measure of sleep disruption than conventional sleep macro-architecture [14,16]. This method is more sensitive for identifying the impact of SDB on sleep quality than simply measuring sleep macro-architecture in 30 s epochs throughout the night [14]. Previously, we identified that children with DS had greater hypoxic exposure and more respiratory events during REM sleep, as well as higher total, delta, sigma, and beta EEG power in REM than typically developing children matched for OSA severity, suggesting that OSA has a greater impact on sleep quality in children with DS compared to typically developing children [17].

In typically developing children, the most common cause of SDB is adenotonsillar hypertrophy, and the first line of treatment for most children with OSA is adenotonsillectomy (AT) [2], which has an approximately 80% success rate for treating OSA [18]. In typically developing children, adenotonsillectomy improves SDB severity, sleep macro-architecture, behavior, and quality of life [19]. In children with DS, in addition to adenotonsillar hypertrophy, the craniofacial and neuromuscular characteristics of the condition mean that SDB is multifactorial, with multiple potential sites of obstruction. In these children, AT is not always effective in resolving SDB, with between 12 and 48% of children still having residual SDB after surgery [20] and requiring additional treatment. Continuous positive airway pressure (CPAP) is commonly used as a second line of treatment, together with other surgical options such as removal of lingual tonsils, adenoidal regrowth, tongue reduction, and uvulopalatopharyngoplasty [21].

In this study, our primary aim was to identify if treatment of SDB would improve sleep macro-architecture, and our secondary aim was to identify if sleep micro-architecture, which is a more sensitive measure of sleep quality, would be improved. We also aimed to identify if any improvement in sleep quality would be associated with improvements in parental reports of sleep, behavior, daytime functioning, and quality of life.

## 2. Methods

Ethical approval for this study was provided from the Monash University and Monash Health Human Research Ethics Committees (15048A) on 15 August 2018. Written informed consent was obtained from parents and verbal assent from children aged over 7 years. No monetary incentive was provided for participation.

### 2.1. Subjects

Children with DS aged 3–19 years referred for assessment of SDB were recruited between May 2016 and March 2018. Parents were asked to participate in a follow-up study with an identical protocol to assess the effects of treatment approximately 2 years after the initial study, with children returning between August 2018 and May 2021. A number of follow-up studies had to be delayed due to the closure of the sleep laboratory for research subjects between March and December 2020 as a result of the COVID-19 pandemic. Parents completed a medical history form and demographic questionnaire. Children with DS were well at the time of the PSG studies.

### 2.2. Protocol

All children underwent an overnight-attended PSG using standard pediatric recording techniques [22]. Prior to the PSG study, height and weight were measured and body mass index (BMI) z-score was calculated. Obesity was defined as ≥95th percentile (BMI z-scores ≥ 1.65) and overweight as ≥85th percentile (BMI z-scores ≥ 1.04) [23]. Parents also completed a number of questionnaires to assess health related quality of life (QOL), behavior, daytime functioning, and sleep problems as detailed below.

The OSA-18 assesses QOL in children with OSA [24]. It comprises 18 questions categorized under five domains: sleep disturbance, physical symptoms, emotional symptoms, daytime function, and caregiver concerns. Scores are rated using a Likert scale to gather information about how frequently the child experienced symptoms in the previous four weeks, with scores ranging from “1—None of the time” to “7—All of the time”. Three threshold levels represent the impact of OSA on QOL, with scores <60 suggesting a small impact, scores between 60 and 80 suggesting a moderate impact, and scores >80 suggesting a significant impact on QOL [25].

The Child Behavior Checklist (CBCL) [26] targeted for children aged between 1.5 and 5 years old contains 99 questions, and for children 6–18 years old contains 118 questions. Scales include internalizing and externalizing behavior and total problems, with the sub-scales in both age groups being equivalent [27]. Questions are rated according to a Likert scale as either 0 (not true), 1 (somewhat or sometimes true), or 2 (very true or often true). The raw scores were calculated by tallying the items under each scale and then converting them to normal-referenced T-scores. T-scores ≥ 70, which are equivalent to scores above the 98th percentile, are considered to be of clinical concern [28].

The Adaptive Behavior Assessment System II (ABAS—Second Edition) [29] was used to measure skills of daily functioning divided into three major adaptive domains including conceptual, social, and practical tasks, covering eleven skill areas. Separate versions of the questionnaire were used for children aged from 0 to 5 years and those aged between 5 and 21 years. The subdomains assessed under the conceptual domain include communication, functional academics, and self-direction; under the social domain are leisure and social skills, and the practical domain assesses community use, home and school living, health and safety, self-care, work (if the child holds a part-time or full-time job), and motor skills. Each skill area is scored according to a four-point Likert scale pertaining to the frequency at which each activity is performed, with scores rated as 0 (is not able to), 1 (never when needed), 2 (sometimes when needed), or 3 (always when needed). The general adaptive composite is a composite score of all adaptive skill areas. Scaled scores for the three adaptive domains and general adaptive composite scores are classified as very superior (130 or more), superior (120–129), above average (110–119), average (90–109), below average (80–89), low (71–79), and extremely low (70 or less) [29].

The Pediatric Sleep Survey Instrument (PSSI) assesses a range of sleep related disorders in children [30]. It contains 26 questions and items rated according to a four-point Likert scale of “Never”, “Rarely—once a week”, “Sometimes—2 to 4 times a week”, or “Usually—5 to 7 times a week”. The items are summed and scores are assigned to assess sleep routine, bedtime anxiety, morning tiredness, night arousals, sleep disordered breathing (SDB), and restless sleep. Raw scores were converted to T-scores with a T-score of >70, which is equivalent to a score above the 95th percentile, being indicative of clinical concern [30].

The Epworth Sleepiness Scale adapted for children and adolescents (ESS-CHAD) is a measure of daytime sleepiness [31]. To date, this questionnaire is validated as a reliable measure of self-reported daytime sleepiness in TD adolescents between the ages of 12 and 18 years old [31] and has been subsequently used in children with Prader–Willi syndrome who are also at an increased risk of SDB [32]. Parents completed the questionnaire in this study on behalf of their child for eight common daily situations, which were scored by how likely their child was to doze or fall asleep during each situation on a scale of 0 (would never doze or sleep) to 3 (high chance of dozing or sleeping). The total scores were summed and a score of 10 or higher suggested excessive daytime sleepiness [31].

Electrophysiological signals were recorded using a commercially available PSG system (E-Series or Grael, Compumedics, Melbourne, Australia). The montage included electroencephalogram (EEG) (Cz, F3-M2, F4-M1, C3-M2, C4-M1, O1-M2, O2-M1), right and left electrooculogram (EOG), submental electromyogram (EMG), left and right anterior tibialis muscle EMG, and electrocardiogram (ECG). Respiratory characteristics were captured using abdominal and thoracic respiratory plethysmography (Pro-Tech zRIP ™ Effort Sensor, Pro-Tech Services Inc., Mukilteo, WA, USA), oronasal thermistor, nasal pressure, and transcutaneous carbon dioxide (TcCO_2_), (TCM4/40, Radiometer, Denmark, Copenhagen or Sentec, Therwil, Switzerland). Peripheral oxygen saturation (SpO_2_) was measured using a Bitmos GmbH (Bitmos, Dusseldorf, Germany), which uses Masimo signal extraction technology for signal processing, or a Masimo Radical-7 (Masimo, Irving, CA, USA), with both devices set to a 2-s averaging time.

### 2.3. Sleep and Respiratory Analysis

All PSG studies were scored manually in 30 s epochs for sleep stages (N1, N2, N3, and REM); respiratory events >2 breaths in duration and arousals were scored by trained pediatric sleep scientists using Compumedics ProFusion software according to American Academy of Sleep Medicine pediatric guidelines [22]. An obstructive apnea was defined as the cessation of airflow in association with ongoing respiratory effort; an obstructive hypopnea was defined as ≥30% decrease in nasal pressure signal amplitude, associated with an increase in labored breathing and an arousal or ≥3% decrease in oxygen saturation; a central apnea was defined as cessation of airflow without inspiratory effort lasting either ≥20 s or at least the duration of two breaths and associated with an arousal or ≥3% oxygen desaturation; and a central hypopnea as ≥30% decrease in nasal pressure signal amplitude with reduced inspiratory effort throughout the entire duration of the event. A mixed apnea was defined if an event was associated with absent respiratory effort during one portion of the event and the presence of obstructed inspiratory efforts in another portion, regardless of which portion came first [22]. The obstructive apnea hypopnea index (OAHI), defined as the total number of obstructive apneas, mixed apneas, and obstructive hypopneas per hour of total sleep time (TST) was used to define SDB severity. Other respiratory parameters included the respiratory disturbance index (RDI), defined as the total number of respiratory events including obstructive and central apneas, mixed apneas, obstructive and central hypopneas, and respiratory event related arousals; the REM RDI; the arousal index (ArI), defined as the number of cortical EEG arousals per hour of TST; and the central apnea hypopnea index (CnAHI), defined as the number of central apneas and hypopneas per hour of TST. Desaturation measures included the average SpO_2_ drop, defined as the average SpO_2_ desaturation with scored respiratory events; SpO_2_ nadir, the lowest oxygen saturation associated with a respiratory event; ODI 4%, defined as the number of times the SpO_2_ dropped by ≥4% per hour of TST; ODI 90%, defined as the number of times the SpO_2_ dropped below 90% per hour of TST; and the average transcutaneous pCO_2_ during TST (Av TcCO_2_). SDB severity categories were based on the obstructive apnea hypopnea index (OAHI) (primary snoring (PS) was defined as an OAHI ≤1 events/h, mild OSA as an OAHI of >1–≤5 events/h, moderate OSA as an OAHI of >5–≤10 events/h, and severe OSA as >10 events/h). Improvement in SDB severity was defined as moving to a less severe SDB category.

### 2.4. Sleep Macro-Architecture Analysis

Standard clinical measures of sleep quality were calculated for each participant and included the following parameters: The duration of each sleep stage (N1, N2, N3, REM) was expressed as a % of TST. Wake after sleep onset (WASO) was calculated as the percentage of time awake during the sleep period time (SPT), defined as the amount of time in minutes from sleep onset until the lights went on at the end of the study, including all periods of wake in between. TST was defined as SPT excluding all periods of wake. Time in bed (TIB) was defined as the time between lights off and lights on. Sleep latency was defined as the period from lights off to the first 3 consecutive epochs of N1 sleep or an epoch of any other stage. Sleep efficiency was defined as the ratio of TST to TIB.

### 2.5. Sleep Micro-Architecture—Spectral Analysis

Micro-architecture was assessed using spectral analysis of the EEG signal performed in Labchart 7.2 (ADInstruments, Sydney, Australia), as described previously [17]. Raw EEG signals were recorded using a band-pass filter of 0.3 Hz to 100 Hz and a sampling frequency of 512 Hz. Spectral analysis was performed on two EEG channels, C4-M1 and F4-M1. To remove any low or high frequency artifact from the signal, the EEG signal was digitally re-filtered using a band-pass filter of 0.5 Hz to 30 Hz. Epochs containing significant artifact, defined as a 30 s epoch containing >10 s of movement artifact that interrupted the EEG signal, were manually excluded from analysis. Frequency bands were set as: delta power (0.5–3.9 Hz), theta power (4–7.9 Hz), alpha power (8–11.9 Hz), sigma power (12–13.9 Hz), and beta power (14–30 Hz) 4. Spectral edge frequency (SEF) was calculated as the frequency below which 90% of EEG power was present. Spectral analysis was run using a fast Fourier transform (FFT) size of 1024 over the entire PSG recording with a Hanning window, which allowed edge effects to be avoided. The FFT output provided total power in 2 s blocks with a frequency resolution of 0.5 Hz. These 0.5 Hz frequency bins were subsequently summed within the 5 frequency bands, producing a single power value for each band. In addition, total power for each 2 s block was determined (0.5–30 Hz). A mean value for each frequency was calculated for each 30 s epoch then averaged per sleep stage within each child.

### 2.6. Statistical Analysis

Statistical analysis was performed using SigmaPlot (SigmaPlot Version 14.5, Systat Software, San Jose, CA, USA). Data were first tested for normality and equal variance. Baseline demographic, sleep, and respiratory data were compared between those children who did and did not participate in the follow-up study with a Mann–Whitney rank sum test. Data are reported as median and interquartile range (IQR). Data for demographic, sleep macro-architecture, respiratory, and EEG spectral parameters were compared between baseline and follow-up studies, both in children who received treatment and those who did not, using two-way repeated measures analysis of variance (RMANOVA) with Bonferroni post-hoc analysis if differences were identified, as the majority of data were parametric. These data are presented as mean ± standard deviation (SD). A *p* value < 0.05 was considered statistically significant.

## 3. Results

Forty-four children completed a PSG study at baseline. Two children had undergone surgery for non-cyanotic congenital heart disease as infants but were considered to have no active cardiac diseases at the time of the studies, four had undergone an adenotonsillectomy prior to the baseline study, six were on thyroxine for hypothyroidism, and one was taking melatonin for sleep-onset insomnia. Thirty-four (77%) parents agreed to a follow-up study. Six parents could not be contacted and four declined to participate. Of the 34 children who returned, only 24 underwent a PSG study with the remaining 10 children completing only the questionnaires and one week of actigraphy at home (Figure 1). The majority of parents who did not agree to the PSG study stated that their child was diagnosed with PS at baseline and they did not wish them to undergo another PSG study as they felt their SDB had improved or was unchanged.

The demographic and respiratory characteristics of the children at baseline who participated in a follow-up sleep study and those who did not are presented in Table 1. The children who participated in the follow-up study were older, and their hip circumferences were higher (*p* < 0.05 for both). All other demographic and respiratory characteristics were similar between the groups, with the exception of average TcCO_2_ TST, which was higher in the children who underwent a follow-up study (*p* < 0.05). There were no differences in sleep macro-architecture characteristics (data not shown).

Table 2 details the treatment before and after the baseline study as well as the SDB severity category and OAHI at both studies. Following the baseline study, nine (38%) children were treated (six with severe OSA, one with moderate OSA, and two with mild OSA), five with adenotonsillectomy, one with lingual tonsillectomy, one with tonsillectomy, and two with continuous positive airway pressure (CPAP). Following treatment, five (56%) children (two with adenotonsillectomy, one with tonsillectomy, and two with CPAP) had improved SDB, as indicated by their being moved to a less severe SDB severity group. Three children who were treated with adenotonsillectomy/lingual tonsillectomy remained in the severe OSA category.

Of the 15 (62%) untreated children, 5 (33%) improved spontaneously, with 1 moving from moderate to mild OSA and 4 moving from mild OSA to PS, and 6 (40%) remained unchanged (1 moderate OSA, 4 mild OSA and 1 PS). Four children (27%) in the untreated group had worsened SDB severity, with three moving from PS to mild OSA and one from moderate to severe OSA.

### 3.1. The Impact of Treatment of OSA on Sleep Macro-Architecture and Respiratory Parameters

Fifteen (62%) children did not receive treatment after the baseline study and nine (38%) did. The mean time between studies was 22 ± 7 months (range 5–37 months), with no difference between those who were treated (20 ± 9 months) compared to those who were untreated (23 ± 5 months). In the treated group, the time between surgery and the follow-up study was 17 ± 15 months (range 3–33 months).

By design, children were older at the follow-up study (treated: 13.9 ± 3.9 (range 7.1–19.0) years; untreated: 11.0 ± 4.0 (range 4.8–21.7) years) compared to the baseline study (treated: 12.1 ± 4.3 (range 5.6–17.1) years; untreated: 9.1 ± 3.0 (range 3.9–19.1) years). Children who were treated were older at both baseline and follow-up studies (*p* < 0.001). In the treated group, five children were obese (56%) and two were overweight (22%) at baseline, compared to four (27%) that were obese in the untreated group. The BMI z-score was higher at baseline in the treated group compared to the untreated group (1.68 ± 0.66 vs. 0.82 ± 0.73, *p* < 0.01) and follow-up (1.83 ± 0.62 vs. 1.03 ± 0.70, *p* < 0.05). There were no differences in BMI z-scores between studies in either group.

Sleep and respiratory data are compared between the untreated and treated groups at baseline and follow-up in Table 3.

At baseline, there was no difference between the untreated and treated groups for time in bed, sleep period time, sleep latency, percent time spent in N1, N2, N3, NREM, or REM sleep, CAHI, average SpO_2_ drop, or average TcCO_2_ TST. Total sleep time and sleep efficiency were lower, and wake after sleep onset as well as REM latency were higher in the treated group. SDB severity was greater in the treated group as expected, with higher OAHI, RDI, REM RDI, ArI, and SpO_2_ > 4% dips/h, and lower SpO_2_ nadir.

At follow-up, there were no differences between the treated and untreated groups for any of the sleep macro-architecture parameters. There were no differences in any of the sleep macro-architecture parameters in the untreated group between baseline and follow-up, however WASO was lower in the treated group at follow-up compared to baseline (*p* < 0.01). SDB severity improved in the treated group with lower OAHI, RDI, ArI, and SpO_2_ > 4% dips/h, and higher SpO_2_ nadir compared to baseline. In the untreated group, there was no improvement in measures of SDB severity at follow-up, however the average SpO_2_ drop was higher at follow-up compared to baseline. Individual data for changes in RDI and OAHI are presented in Figure 2.

### 3.2. The Impact of Treatment of OSA on Sleep Micro-Architecture

C4-derived spectral power in N2, N3, and REM at baseline and follow-up in the untreated and treated groups are presented in Table 4. The treated group had lower SEF in N3 at baseline compared to follow-up. The untreated group had lower SEF in N2 and REM at baseline compared to follow-up. Theta power was higher in N2 and N3 at baseline than follow-up. In REM, total, delta, theta, and alpha power were higher and SEF was lower at baseline compared to follow-up. When the groups were compared, theta and alpha powers in N2 and REM were higher in the untreated compared to the treated group at baseline.

### 3.3. The Impact of Treatment of OSA on Behavior, Daytime Functioning, and Quality of Life

Behavioral, daytime functioning, quality of life, and parental sleep questionnaire data in the untreated and treated groups at baseline and follow-up are presented in Table 5. There were no differences in internalizing, externalizing, or total behavioral problems as reported in the CBCL or any of the sub-scores of the ABAS between the untreated and treated groups at either baseline or follow-up. The OSA-18 physical symptom score improved in the untreated group between baseline and follow-up (*p* = 0.037) and tended to improve in the treated group (*p* = 0.091). The total score on the OSA-18 was significantly lower at follow-up in the treated group (*p* < 0.05), indicating improved quality of life. At baseline, two children in each group had total OSA-18 scores >80, indicating a significant impact of SDB on QOL; however, at follow-up, there were no scores >80 in either group. The SDB sub-scale of the PSSI was higher in the treated group at baseline compared to the untreated group (*p* < 0.01), but there were no differences at follow-up either between groups or studies. Although there were no differences in the ESS-CHAD mean scores between groups, two children scored >10, indicating excessive daytime sleepiness at follow-up in the untreated group compared to no children scoring in this range in the treated group.

## 4. Discussion

SDB is very common in children with DS, however there are few studies which have investigated the effects of treatment on sleep quality, daytime performance, and QOL. As expected, at baseline the treated children had more severe SDB than the children who were not treated. SDB severity significantly improved in the treated children and worsened in some untreated children. However, conventional sleep macro-architecture measures did not improve between baseline and follow-up studies in the treated children. This was the first study to use spectral analysis of the EEG to assess changes in sleep micro-architecture following treatment of SDB in children with DS. Surprisingly, there were minimal changes in sleep micro-architecture between studies in the treated group, with most differences being observed in the untreated group. Quality of life improved in the treated group, with the OSA-18 total score being significantly lower at follow-up in the treated group (*p* < 0.05). The PSSI SDB scale also tended to improve in the treated group, suggesting that improvement in QOL may be related to improved SDB symptoms.

As would be expected clinically, the group of children who were treated had more severe SDB than those who were untreated, with almost half (47%) of the untreated children having undergone previous treatment and having PS or mild OSA at baseline. Although the children who were treated had more severe SDB at baseline, at follow-up, the OAHI overall significantly decreased by around 50% in the treated group from 40.3 ± 46.9 to 17.9 ± 26.9 events/h (*p* < 0.01). In contrast, OAHI in the untreated group increased slightly from 4.3 ± 5.7 to 7.3 ± 14.7 events/h, indicating that in some children SDB had worsened (as shown in Figure 2). Our findings of no significant effects on sleep quality, behavior, and daytime functioning were likely affected by the fact that the majority of children in both the treated (78%) and untreated (73%) groups still had residual OSA at follow-up. OSA has been demonstrated to affect daytime behavior in children with DS compared to no OSA [8], however not all studies show this [9].

In the treated group, seven children had severe OSA, one had moderate OSA, and one had mild OSA at baseline. None of the children received treatment prior to the baseline study. In the severe group, two children were treated with CPAP, and both showed significant improvement at the follow-up study. The remaining five children had surgical treatment (three adenotonsillectomy, one tonsillectomy, and one lingual tonsillectomy). At follow-up in the surgically treated children, OAHI improved in two children, but they still had severe OSA, and severity worsened in three. Our study confirms previous findings that also identified that, although treatment improves OSA severity in children with DS, a significant proportion have residual OSA and further treatments are often required [33].

In the untreated group, seven children previously underwent surgical treatment (six adenotonsillectomy and one tonsillectomy). At follow-up, SDB severity worsened in five (33%) of the untreated children, remained unchanged in six (40%), and improved in four (27%), suggesting that regular assessment is required to maximize health and QOL in these children. Previous studies in typically developing children have also demonstrated that mild SDB can improve spontaneously over time. In the childhood adenotonsillectomy trial (CHAT) study, 46% of children aged 5–9 years resolved spontaneously, defined as an apnea hypopnea index < 2 events/h in 7 months, however this was significantly fewer than in the treated group, where 79% resolved [19].

Previous studies in typically developing children also show that surgical treatment of SDB is not always effective in resolving the disorder. A previous large multicenter study of 578 children (mean age, 6.9 ± 3.8 year), of which approximately 50% were obese, demonstrated that although AT resulted in a significant AHI reduction from 18.2 ± 21.4 to 4.1 ± 6.4 events/h (*p* < 0.001), only 27.2% had complete resolution of OSA, defined as an AHI <1 event/h) [34]. Age and body mass index z-score were the two principal factors contributing to post-AT AHI (*p* < 0.001). In the current study, 78% of the treated group were overweight or obese compared to 27% in the untreated group at baseline, and BMI z-score did not change in either the untreated or treated groups between studies. In the children who were treated surgically (*n* = 7), obesity did not seem to play a role in improvement of SDB severity, with one normal-weight child improving and one worsening. In the overweight/obese children, two improved and three did not. However, our small numbers do not allow us to make any firm conclusions about the relationship between treatment success and BMI in this study. Studies of the effectiveness of treatment for OSA in children with DS demonstrate a significantly reduced chance of cure following AT compared to typically developing children, with one study reporting 73% of children with DS required CPAP, bi-level PAP, or supplementary oxygen for persistent OSA after AT [35]. In a study of 45 children with DS examined before and after surgery, the median OAHI decreased from 9.3 events/h (range 0.2–74.4 events/h) to 3.4 events/h (range 0.4–37.7 events/h) [36]. The study, however, did not report how many children still had residual OSA. A meta-analysis of five studies including 118 children demonstrated that although AT reduced SDB severity in many cases, it was noncurative, with up to 75% needing postoperative breathing support, in addition to a high rate of immediate postoperative airway complications, and with no change in sleep efficiency or architecture [33]. The papers included consistently reported moderate success in improving polysomnographic parameters, and limited pooling of the data demonstrated a mean decrease of the apnea–hypopnea index by 51% (95% confidence interval [CI], 46–55%) [33]. Findings are similar to the improvement in OAHI in the treated group in our study. A recent study of 33 children with DS who were followed up ≤1 year after surgery reported that only 15.2% of the children had normal postoperative AHI values and 63.6% still had moderate to severe OSA [37].

Our study showed that conventional sleep macro-architecture was not different between the baseline and follow-up in either the treated or untreated groups. These findings are supported by previous studies in children with DS, which also showed that although SDB severity improved, there were minimal changes in sleep macro-architecture [36]. Spectral analysis of the EEG provides a more sensitive measure of sleep disruption, but contrary to our hypothesis, we did not identify any changes in sleep micro-architecture following treatment. This finding is likely due to the residual OSA, which was still apparent in 67% of the children following treatment. Conversely, it could also be that SDB does not have a significant impact on conventional sleep quality measures in children with DS. Previously, we showed that there were few differences in sleep macro-architecture in children with DS compared to typically developing children with and without SDB [17]. This is supported by studies of typically developing children, which also demonstrate that sleep macro-architecture is not significantly altered in children with SDB [14].

In our previous study, where we compared EEG spectral power in some of these same children at baseline to that of typically developing children with and without SDB, we identified elevated theta power in REM sleep in children with DS compared to the typically developing children without SDB [17]. This suggests that REM sleep is more disrupted in children with DS, a finding that may be related to the increased number of respiratory events in REM sleep for this group. In the current study, we did not identify any significant effects of treatment on sleep micro-architecture. We did, however, identify that at baseline the treated group had lower theta power in N2 and REM sleep and lower total power in REM sleep compared to the untreated group. These findings are difficult to explain because increased theta power is related to arousal; therefore, it would be expected that the treated group who had more severe OSA and higher arousal indices at baseline than the untreated group would have elevated rather than lower theta power.

In the untreated group, during N2, N3, and REM sleep, theta power was higher at baseline compared to follow-up, suggesting that sleep was more disrupted at baseline compared to follow-up in the untreated children, despite no significant change in SDB severity. In addition, total, delta, and alpha power were higher in REM sleep at baseline compared to follow-up in the untreated group. Higher delta power in REM sleep was previously reported in adult patients with OSA compared to healthy controls [38]. These authors suggested that slow EEG activity may be a sign of arousal in response to obstructive apneas. Previously, we have shown that children with DS have more respiratory events in REM sleep compared to age and SDB severity-matched typically developing children [17]. Thus, the increased delta power in REM sleep in the untreated children at baseline may also be related to the increased number of respiratory events, and therefore respiratory arousals, observed in REM sleep in these children.

Despite improvement in SDB severity in the treated group, we identified no changes in behavior or daytime functioning. This finding is in contrast to a previous small study of six children with DS, who were included in a sub-sample of ten children with neurodevelopmental disability that analyzed the effect of positive airway pressure (PAP) therapy on neurobehavioral outcomes in a heterogeneous group of 52 children aged 2–16 years with OSA [39]. Following three months of PAP use, the children with developmental delay experienced significant improvements in daytime sleepiness, internalizing, and total behavior scores. The authors noted however, that the study was underpowered for the developmentally delayed children and changes in other behavioral parameters were potentially missed. The difference in findings may be due to the number of children with residual SDB in our study, where only two children were on CPAP therapy compared to the study by Marcus et al., [39]. It may also be due to the tests administered not being sensitive enough to detect any changes in this group of children. We noted that parental responses were very varied, with two parents in the treated group reporting a ≥10 point deterioration in the total CBCL score between the two studies, one parent reporting an improvement of 20 points, and the remaining parents reporting either no change or a very small change. Similarly, in the untreated group one parent reported a large improvement, while another parent reported a large worsening. The majority of parents reported very little change. There was similar variability in the ABAS scores. Although the variability of parental reports is of concern, it was recently reported that the CBCL can be used as a screening measure when evaluating behavioral concerns among children with DS, but that it has poor discriminant validity, and key behavior concerns in DS may not be captured by the CBCL [40]. In addition, factors other than OSA may impact daytime behavior and functioning in this group of children. Children with DS are also at increased risk of behavioral sleep disorders (see [1] for review), which may affect sleep quality and daytime functioning; however, in the current study, we did not identify any scores on the PSSI that were >70, with the exception of the SDB sub-scale, which would be of clinical concern. We did, however, identify that parents reported improved QOL in the treated group, which was not identified in the untreated group. This was previously reported in typically developing children who underwent AT compared to those who were in the watchful waiting group, despite improvements in disease severity in both groups [41]. Parents reported improved physical symptoms on the OSA-18 in both groups and this reached statistical significance in the untreated group. This could be due to a mismatch between subjective and objective measures, which we previously reported in relation to assessments of working memory in typically developing children with SDB [42]. Improved QOL following CPAP treatment [39,43] and also following AT has previously been reported [43].

We must acknowledge the limitations of our study. Firstly, COVID-19 lockdowns delayed many of the follow-up studies and potentially also impacted parents’ or children’s willingness to return for an overnight sleep study. Distress related to the process of the sleep study may also impact sleep quality, making comparison of two PSG results more difficult; however, we did not identify any differences in sleep duration between the two studies. Our sample size (*n* = 24) at follow-up was small, however, similar studies also included similar numbers of children with DS ([8,9,11,44], which reflects the difficulty of recruitment for the study, as it involved considerable time commitment for parents and a complex medical procedure for children. We did not control for the effect of treatment prior to the baseline study, and potentially affected the group comparisons. Children in our study were typical of those referred to our clinic, and treatment decisions were made based on clinical history and parent preferences. Further studies are required to identify if surgical or medical treatment of milder SDB also improves severity.

In conclusion, our study identified that treatment of SDB improves severity of the disease as defined by PSG parameters. This was associated with parental reports of improved QOL, despite treatment having no demonstrable impacts on sleep quality, behavior, or daytime functioning. It must be acknowledged that our sample size was small, and the majority of children in both the treated and untreated groups had residual OSA, necessitating the need for further, larger studies to confirm our findings.

## Figures and Tables

**Figure 1 children-09-00984-f001:**
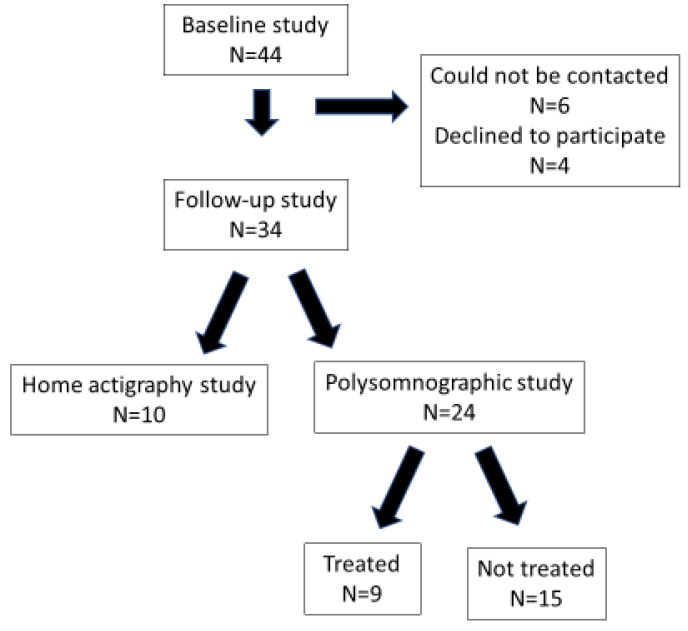
Flow diagram illustrating subject recruitment to the follow-up study.

**Figure 2 children-09-00984-f002:**
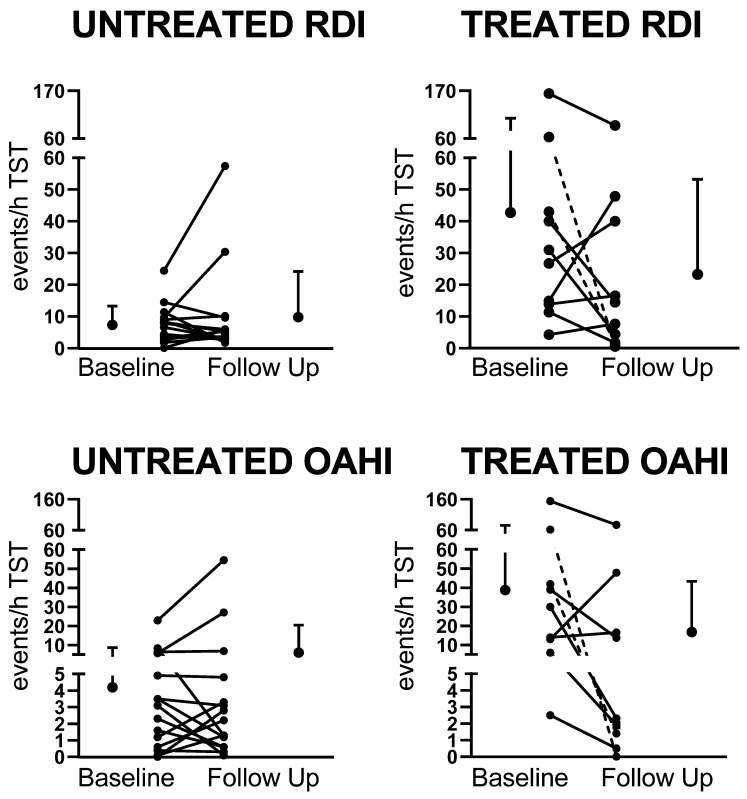
Changes in respiratory disturbance index (RDI) and obstructive apnea hypopnea index (OAHI) between baseline and follow-up in the untreated and treated groups. Dotted lines indicate the two children treated with continuous positive airway pressure (CPAP).

**Table 1 children-09-00984-t001:** Demographic and respiratory characteristics at baseline of children with Down syndrome who participated in a follow-up PSG study and those who did not. Data are reported as median and IQR.

	No Follow-Up PSG (*n* = 20)	Follow-Up PSG (*n* = 24)	*p*-Value
Females/Males	11/9	13/11	NS
Age (years)	5.5 (4.2, 12.0)	8.8 (6.8, 14.5)	<0.05
BMI z-score	0.8 (0.1, 2.1)	1.1 (0.6, 1.8)	NS
Neck circumference (cm)	28.5 (26.3, 33.0)	31.0 (28.0, 38.0)	NS
Waist circumference (cm)	59.0 (53.0, 72.0)	64.0 (57.0, 80.0)	NS
Hip circumference (cm)	58.0 (56.0, 77.0)	72.0 (64.0, 90.0)	<0.05
OAHI (events/h)	1.8 (0.2, 9.0)	5.3 (1.8, 20.7)	NS
RDI (events/h)	4.9 (2.8, 12.8)	9.4 (4.3, 26.1)	NS
REM RDI (events/h)	13.6 (5.3, 24.9)	19.1 (9.4, 31.6)	NS
CAHI (events/h)	1.2 (0.4, 2.6)	1.8 (1.5, 4.0)	NS
SpO_2_ nadir (%)	89.0 (85.3, 90.8)	88.0 (86.0, 89.8)	NS
Average SpO_2_ drop	4.0 (3.0, 5.0)	4.0 (3.0, 5.0)	NS
SpO_2_ < 90%/h	0.1 (0.0, 0.6)	0.2 (0.1, 0.4)	NS
SpO_2_ > 4% drop/h	2.0 (1.0, 5.0)	3.9 (1.7, 8.0)	NS
Arousal index (events/h)	12.0 (9.7, 19.2)	14.7 (13.0, 18.2)	NS
Average TcCO_2_ TST	42.6 (40.1, 46.8)	48.0 (43.7, 49.9)	<0.05

OAHI, obstructive apnea hypopnea index; CAHI, central apnea hypopnea index; RDI, respiratory disturbance index; REM, rapid eye movement; TST, and total sleeping time.

**Table 2 children-09-00984-t002:** Sleep disordered breathing severity and type of treatment prior to the baseline study and prior to the follow-up study.

Treatment and SDB Severity Characteristics				
Treatment before Baseline Study	Baseline SDB Severity Group (OAHI)	Treatment after Baseline Study	Follow Up SDB Severity Group (OAHI)	Improved
**Children Treated at Follow-up**				
No treatment	Severe (13.9)	Adenotonsillectomy	Severe (16.5)	No
No treatment	Severe (154)	Lingual tonsillectomy	Severe (77.0)	No
No treatment	Severe (30.0)	Adenotonsillectomy	Mild (2.3)	Yes
No treatment	Severe (13.0)	Adenotonsillectomy	Severe (47.9)	No
No treatment	Severe (39.0)	Adenotonsillectomy	Severe (13.7)	No
No treatment	Severe (62.0)	CPAP	PS (0.0)	Yes
No treatment	Severe (42.0)	CPAP	Mild (1.4)	Yes
No treatment	Mod (5.9)	Tonsillectomy	Mild (1.9)	Yes
No treatment	Mild (2.5)	Adenotonsillectomy	PS (0.5)	Yes
**Children Un-treated at Follow-up**				
No treatment	Severe (22.9)	No treatment	Severe (54.6)	No
No treatment	Moderate (5.7)	No treatment	Severe (27.1)	No
Adenotonsillectomy	Moderate (8.3)	No treatment	Mild (1.3)	Yes
Adenotonsillectomy	Moderate (6.4)	No treatment	Moderate (6.8)	No
No treatment	Mild (3.5)	No treatment	Mild (3.1)	No
Adenotonsillectomy	Mild (4.9)	No treatment	Mild (4.8)	No
No treatment	Mild (2.3)	No treatment	PS (0.6)	Yes
Adenotonsillectomy	Mild (1.2)	No treatment	Mild (3.3)	No
Tonsillectomy	Mild (1.6)	No treatment	PS (0.6)	Yes
Adenotonsillectomy	Mild (3.5)	No treatment	Mild (1.2)	No
No treatment	Mild (3.1)	No treatment	PS (0.1)	Yes
Tonsillectomy	PS (0.6)	No treatment	Mild (2.2)	No
No treatment	PS (0.4)	No treatment	PS (0.3)	No
No treatment	PS (0.1)	No treatment	Mild (1.3)	No
Adenotonsillectomy	PS (0.0)	No treatment	Mild (2.8)	No

PS primary snoring, CPAP continuous positive airway pressure. For analysis, children were divided into those who were untreated and treated.

**Table 3 children-09-00984-t003:** Sleep macro-architecture and respiratory data in untreated and treated groups at baseline and follow-up. Data are presented as mean ± SD.

	Untreated		Treated	
	Baseline	Follow-Up	Baseline	Follow-Up
N	15	15	9	9
Time in bed (min)	531 ± 45	509 ± 28	518 ± 22	506 ± 42
Sleep period time (min)	485 ± 59	480 ± 36	465 ± 71	458 ± 54
Total sleep time (min)	444 ± 56	424 ± 46	369 ± 83 **	398 ± 62
Wake after sleep onset (%)	9 ± 6	11 ± 7	20 ± 15 **	13 ± 7 †
Sleep efficiency (%)	84 ± 9	84 ± 9	71 ± 16 **	79 ± 10
Sleep latency (min)	42 ± 48	23 ± 17	35 ± 25	37 ± 27
REM latency (min)	164 ± 63	188 ± 78	223 ± 94 *	307 ± 86
N1 (%)	5 ± 4	6 ± 5	9 ± 7	6 ± 4
N2 (%)	48 ± 9	48 ± 6	50 ± 8	49 ± 10
N3 (%)	31 ± 17	30 ± 6	28 ± 10	29 ± 5
NREM (%)	81 ± 6	84 ± 6	87 ± 8	84 ± 6
REM (%)	19 ± 6	16 ± 5	13 ± 8	16 ± 6
OAHI (events/h)	4.3 ± 5.7	7.3 ± 14.7	40.3 ± 46.9 **	17.9 ±26.9 ††
RDI (events/h)	7.7 ± 6.1	10.2 ± 14.8	44.6 ± 48.2 **	24.8 ± 29.6 †
REM RDI (events/h)	14.1 ± 10.2	19.8 ± 22.9	56.0 ± 50.7 *	47.5 ± 65.0
CAHI (events/h)	2.5 ± 1.4	2.3 ± 0.9	4.3 ± 6.9	6.4 ± 12.2
Arousal index (events/h)	13.9 ± 9.7	14.9 ± 6.9	31.4 ± 28.0 **	20.8 ± 17.0 †
SpO_2_ nadir (%)	88.91± 2.9	86.9 ± 5.0	81.6 ± 10.4 **	87.0 ± 6.3 †
Average SpO_2_ drop	3.9 ± 0.8 #	4.4 ± 0.9	4.9 ± 1.8	4.7 ± 1.7
SpO_2_ < 90%/h	0.2 ± 0.3	0.5 ± 0.6	10.2 ± 25.6	4.0 ± 10.2
SpO_2_ > 4% drop/h	2.9 ± 2.1	5.2 ± 7.1	25.7 ± 41.2 *	14.1 ± 23.1 †
Average TcCO_2_ TST	46.2 ± 5.3	42.9 ± 4.4	47.4 ± 4.4	43.3 ± 3.8

REM, rapid eye movement; N1, NREM stage 1; N2, NREM stage 2; N3, NREM stage 3; NREM, non-rapid eye movement, and TST total sleep time. Data presented as mean ± SD; * *p* < 0.05, ** *p* < 0.01 baseline untreated compared to baseline treated; † *p* < 0.05, †† *p* < 0.01 baseline treated compared with follow-up treated; ^#^
*p* < 0.05, baseline untreated compared to follow-up untreated.

**Table 4 children-09-00984-t004:** C-4 derived EEG spectral analysis data in the untreated and treated groups at baseline and follow-up. Data are presented as mean ± SD.

	Untreated		Treated	
	Baseline	Follow-Up	Baseline	Follow-Up
**N2**				
Total Power (µV^2^)	905.2 ± 377.2	673.8 ± 183.4	671.5 ± 395.3	783.9 ± 598.1
SEF (µV^2^)	10.0 ± 2.4 ###	11.2 ± 2.1	10.0 ± 2.1	10.8 ± 1.7
Delta Power (µV^2^)	677.3 ± 284.0	501.9 ± 135.9	538.5 ± 346.6	638.7 ± 506.6
Theta Power (µV^2^)	159.2 ± 87.8 ###	106.9 ± 55.9	91.8 ± 60.4 *	89.3 ± 70.4
Alpha Power (µV^2^)	27.5 ± 10.5	24.5 ± 8.4	16.9 ± 6.0 *	22.0 ± 13.8
Sigma Power (µV^2^)	13.9 ± 8.9	14.9 ± 9.1	6.8 ± 3.2	10.3 ± 6.3
Beta Power (µV^2^)	18.8 ± 8.4	19.6 ± 8.7	12.4 ± 4.8	17.9 ± 13.0
**N3**				
Total Power (µV^2^)	4274.3 ± 1541.5	3740.4 ± 972.5	3820.0 ± 1864.6	3348.5 ± 2859.8
SEF (µV^2^)	5.3 ± 0.6	5.3 ± 0.6	4.9 ± 0.5	5.5 ± 1.1 †
Delta Power (µV^2^)	3920.5 ± 1401.4	3453.0 ± 901.7	3538.1 ± 1712.0	3090.3 ± 2650.1
Theta Power (µV^2^)	283.3 ± 122.4 #	223.1 ± 70.6	229.6 ± 131.8	194.2 ± 157.6
Alpha Power (µV^2^)	34.6 ± 15.9	30.4 ± 9.2	24.5 ± 10.9	25.0 ± 15.1
Sigma Power (µV^2^)	8.4 ± 4.3	8.2 ± 3.8	5.6 ± 2.2	7.2 ± 3.6
Beta Power (µV^2^)	11.1 ± 5.2	12.1 ± 9.9	8.7 ± 3.8	20.0 ± 29.2
**REM**				
Total Power (µV^2^)	799.1 ± 507.4 ##	523.8 ± 254.2	428.5 ± 276.1	510.0 ± 525.8
SEF (µV^2^)	9.5 ± 2.9 ##	10.6 ± 2.7	10.6 ± 2.3	11.4 ± 2.8
Delta Power (µV^2^)	594.5 ± 409.4 ##	383.2 ± 205.1	325.0 ± 219.1	400.4 ± 456.6
Theta Power (µV^2^)	152.8 ± 84.9 ###	97.4 ± 46.3	71.6 ± 47.9 *	75.1 ± 64.3
Alpha Power (µV^2^)	20.4 ± 9.2 ##	16.6 ± 7.3	12.7 ± 5.9 *	13.4 ± 6.3
Sigma Power (µV^2^)	5.1 ± 1.8	5.1 ± 2.1	3.2 ± 1.5	3.8 ± 1.3
Beta Power (µV^2^)	15.5 ± 6.5	15.0 ± 5.7	11.0 ± 5.2	12.1 ± 5.4

* *p* < 0.05, baseline untreated compared to baseline treated; † *p* < 0.05, baseline treated compared with follow-up treated; # *p* < 0.05, ## *p* < 0.01, ### *p* < 0.001 baseline untreated compared to follow-up untreated.

**Table 5 children-09-00984-t005:** Behavioral, daytime functioning, quality of life, and parental sleep questionnaire data in the untreated and treated groups at baseline and follow-up. Data are presented as mean ± SD.

	Untreated		Treated	
	Baseline	Follow-Up	Baseline	Follow-Up
CBCL internalizing problems	55.4 ± 10.4	55.7 ± 10.9	54.3 ± 10.3	56.1 ± 9.4
CBCL externalizing problems	54.6 ± 10.5	57.3 ± 7.7	53.6 ± 7.1	55.0 ± 9.9
CBCL total problems	56.4 ± 10.7	58.2 ± 9.5	55.0 ± 9.5	57.0 ± 8.0
ABAS GAC composite score	55.4 ± 14.1	57.4 ± 12.2	53.8 ± 9.2	53.7 ± 13.1
ABAS conceptual composite score	57.8 ± 11.5	60.4 ± 8.8	55.0 ± 7.0	58.1 ± 10.8
ABAS social composite score	73.1 ± 13.9	73.2 ± 13.2	69.5 ± 11.0	69.2 ± 12.7
ABAS practical composite score	53.0 ± 17.7	54.3 ± 15.1	51.8 ± 15.3	51.2 ± 13.1
OSA-18 sleep disturbances sub-scale	12.2 ± 5.3	11.7 ± 5.7	14.2 ± 5.9	9.7 ± 1.2
OSA-18 physical symptoms sub-scale	13.7 ± 5.6 #	9.1 ± 3.7	14.2 ± 6.9	9.2 ± 3.0
OSA-18 emotional symptoms sub-scale	8.3 ± 4.5	8.2 ± 4.0	10.4 ± 3.1	6.5 ± 4.2
OSA-18 daytime function sub-scale	9.3 ± 4.6	7.4 ± 4.3	9.2 ± 5.2	5.5 ± 1.3
OSA-18 care giver concerns sub-scale	13.0 ± 7.2	11.3 ± 5.8	12.8 ± 8.2	9.0 ± 2.1
OSA-18 total symptoms	56.5 ± 23.0	47.6 ± 17.1	60.8 ± 27.8	39.8 ± 3.8 †
PSSI sleep routine	54.5 ± 10.8	53.4 ± 11.2	51.4 ± 12.5	51.0 ± 6.9
PSSI bed time anxiety	52.2 ± 9.6	54.6 ± 14.4	62.7 ± 15.9	56.0 ± 10.4
PSSI morning tiredness	52.2 ± 13.8	53.2 ± 12.6	52.7 ± 10.5	48.7 ± 7.2
PSSI night arousal	55.5 ± 14.6	59.9 ± 13.7	55.9 ± 17.2	58.0 ± 14.1
PSSI sleep disordered breathing	68.3 ± 13.0	70.9 ± 13.7	78.0 ± 15.2 **	70.2 ± 13.4
PSSI restless sleep	59.8 ± 11.8	55.6 ± 12.8	57.4 ± 6.4	55.3 ± 14.1
ESS-CHAD	4.0 ± 3.9	5.7 ± 4.9	6.0 ± 4.9	3.3 ± 2.3

CBCL Child Behavior Check List; ABAS Adaptive Behavior Assessment System; PSSI Pediatric Sleep Survey Instrument, ESS-CHAD Epworth Sleepiness Scale—child and adolescent; ** *p* <0.01, baseline untreated compared to baseline treated; † *p* < 0.05, baseline treated compared with follow-up treated; # *p* < 0.05, baseline untreated compared to follow-up untreated.

## Data Availability

Data will be available on request.

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
