# Peer review of "Effects of Treatment of Sleep Disordered Breathing on Sleep Macro- and Micro-Architecture in Children with Down Syndrome"

_children, 2022, doi:10.3390/children9070984_

Round 1

Reviewer 1 Report

Major comments:

Overall the paper is interesting, well written and provides us with a detailed assessment of children with Down syndrome post treatment for OSA. The small numbers included in the follow up analysis impact on the interpretation of the findings here. A fuller discussion on some of the papers findings would add to the manuscript.

The methods section is overlong. Although I believe it is important to include all the details on how this cohort was assessed, as this is a paediatric journal and not a sleep journal, I feel much of the details on the sleep analysis can be moved to a supplementary appendix.

Minor comments:

Introduction:

Line 41: “TD children…” please expand on acronym

Methods:

Lines 95-100: Details on baseline characteristics of subjects should be placed in the results and not the methods section.

Results:

Lines 241-247: A flow diagram of patient recruitment and follow up would be helpful to include here to help the reader know how the study group was arrived at.

The results could be presented more clearly in terms of n(%) and denominators. Instead of quoting n, please include n(%) for all results here.

Discussion:

There was a greater % of mild/PS cases and much less % of severe cases of OSA included in the non-treatment group. Could this have influenced your results? The authors need to discuss this.

One of the main limitations is small sample size at follow up. How does this study compare with other similar studies in terms of numbers of children with DS studied.

Please include in the discussion any studies where the CBCL and ABAS have been used in studies on SDB in children with DS.

There was a statistically significant difference in the OSA-18 physical symptom score in the untreated group yet their OAHI increased. Could the authors postulate on why this occurred?

Lines 374 and 383: Add punctuations at beginnings of lines please.

Line 401:  “N” should not be capitalised.

 Line 477: “Clearly, factors other than OSA were impacting on daytime behaviour and functioning”. Behaviour sleep disorders are common in children with DS and can occur in the absence or presence of SDB. Some of this information was captured by the PSSI questionnaire, can the authors discuss as to whether or not behaviour sleep disorders played a role in addition to SDB in these children.

Author Response

Responses to Reviewer 1

Major comments:

Overall the paper is interesting, well written and provides us with a detailed assessment of children with Down syndrome post treatment for OSA. The small numbers included in the follow up analysis impact on the interpretation of the findings here. A fuller discussion on some of the papers findings would add to the manuscript.

See specific sections below 

The methods section is overlong. Although I believe it is important to include all the details on how this cohort was assessed, as this is a paediatric journal and not a sleep journal, I feel much of the details on the sleep analysis can be moved to a supplementary appendix.

We are sorry that the reviewer thinks our methods section is too long however we feel we need to include all the details of the sleep study and the instructions to authors do not provide details of the availability of any supplementary data appendices.

Minor comments:

Introduction:

Line 41: “TD children…” please expand on acronym

TD stands for typically developing and this has been added to the manuscript.

Methods:

Lines 95-100: Details on baseline characteristics of subjects should be placed in the results and not the methods section.

We have moved this section to the results as suggested- now on page 5. 

Results:

Lines 241-247: A flow diagram of patient recruitment and follow up would be helpful to include here to help the reader know how the study group was arrived at.

Thank you for this suggestion. We have added a new figure, which is now Figure 1. 

The results could be presented more clearly in terms of n(%) and denominators. Instead of quoting n, please include n(%) for all results here.

We have added % to the results. 

Discussion:

There was a greater % of mild/PS cases and much less % of severe cases of OSA included in the non-treatment group. Could this have influenced your results? The authors need to discuss this.

Thank you we have added the following to the discussion on page on page 13.

“As would be expected clinically, the group of children who were treated had more severe SDB than those who were untreated, with almost half (47%) of the untreated children having undergone previous treatment and having PS or Mild OSA at baseline. Although the children who were treated had more severe SDB at baseline, at follow-up overall the OAHI had significantly decreased by around 50% in the treated group from 40.3 ± 46.9 to 17.9 ± 26.9 events/h (p<0.01). In contrast, in the untreated group the OAHI had increased slightly from 4.3 ± 5.7 to 7.3 ± 14.7 events/h, indicating that in some children SDB had worsened (as shown in Figure 2). Our findings of no significant effects on sleep quality, behavior and daytime functioning may have been affected by the fact that the majority of children in both the treated (78%) and untreated (73%) groups still had residual OSA at follow-up. OSA has been demonstrated to affect daytime behavior in children with DS compared to no OSA [8], however not all studies have shown this [9].” 

One of the main limitations is small sample size at follow up. How does this study compare with other similar studies in terms of numbers of children with DS studied.

Thank you we have added this on page 16

“Our sample size (n=24) at follow-up was small, however similar studies have also included similar numbers of children with DS [8, 9, 11, 43] which reflects the difficulty of recruitment for a study that involved considerable time commitment for parents and a complex medical procedure for children”. 

Please include in the discussion any studies where the CBCL and ABAS have been used in studies on SDB in children with DS.

We have included this on page 15

“Although the variability of parental report is of concern, it has recently been reported that the CBCL can be used as a screening measure when evaluating behavioral concerns among children with DS, but that it has poor discriminant validity and that key behavior concerns in DS may not be captured by the CBCL [40].” 

There was a statistically significant difference in the OSA-18 physical symptom score in the untreated group yet their OAHI increased. Could the authors postulate on why this occurred?

We have added this on page 15

“We did however identify that parents reported improved QOL in the treated group which was not identified in the untreated group. This has previously been reported in TD children who underwent AT compared to those who were in the watchful waiting group, despite improvements in disease severity in both groups [41]. Parents reported improved physical symptoms on the OSA-18 in both groups and this reached statistical significance in the untreated group. This could be due to a mismatch between subjective and objective measures, which we have previously reported in relation to assessments of working memory in children with SDB [42]. Improved QOL following CPAP treatment [39, 43] and also following AT has previously been reported [43].” 

Lines 374 and 383: Add punctuations at beginnings of lines please.

This has been corrected 

Line 401:  “N” should not be capitalised.

This has been corrected 

 Line 477: “Clearly, factors other than OSA were impacting on daytime behaviour and functioning”. Behaviour sleep disorders are common in children with DS and can occur in the absence or presence of SDB. Some of this information was captured by the PSSI questionnaire, can the authors discuss as to whether or not behaviour sleep disorders played a role in addition to SDB in these children.

Thank you we have expanded our discussion on page 15

“In addition, factors other than OSA may be impacting on daytime behavior and functioning in this group of children. Children with DS are also at increased risk of behavioral sleep disorders (see [1] for review) which may affect sleep quality and daytime functioning, however in the current study we did not identify any scores on the PSSI that were > 70, with the exception of the SDB subscale, which would be of clinical concern”.

Reviewer 2 Report

This study topic is relevant and of interest, addressing an area where current evidence is limited.

Whilst the aim of the paper is stated, I am not if the primary aim of the study was to try to assess the effect of treatment on sleep microstructure (inferring sleep quality) or the effect of treatment on behaviour and function or both. It is also unclear how this is clinically relevant- what would be the benefit of using sleep microarchitecture rather than current PSG parameters for clinicians- both still require a PSG to be undertaken. Is the author suggesting PSG microarchitecture would be used to predict outcome? I think the clinical context is missing. 

Study methodology is clear. The measures used are reasonable overall but i am not sure of the value of using the ESS-CHAD completed by parents in this population of children. 

There is a lot of data in this study. The layout of the results could be improved perhaps with some clear subheadings so the reader can follow which findings are being described. Perhaps a flow diagram of the treatment or non treatment paths from baseline would help readers to follow the findings. 

Whilst the authors acknowledge the limitation of small number of participants in their study they still draw a conclusion about treatment effect based on only 9 treated participants, 4 of whom still had severe OSA after treatment so are actually still not technically treated. I think it is wrong to conclude that there is no effect on sleep quality, behaviour and function from just five patients and I am concerned that the paper would be confusing with an unjustified take home message for an inexperienced reader. 

Author Response

Responses to Reviewer 2

This study topic is relevant and of interest, addressing an area where current evidence is limited.

Whilst the aim of the paper is stated, I am not if the primary aim of the study was to try to assess the effect of treatment on sleep microstructure (inferring sleep quality) or the effect of treatment on behaviour and function or both. It is also unclear how this is clinically relevant- what would be the benefit of using sleep microarchitecture rather than current PSG parameters for clinicians- both still require a PSG to be undertaken. Is the author suggesting PSG microarchitecture would be used to predict outcome? I think the clinical context is missing. 

We have clarified our aims. We are not suggesting that spectral analysis of the EEG to examine sleep micro-architecture should be used clinically. We have used spectral analysis as it is a more sensitive measure of assessing sleep quality and we hoped to identify improvements in sleep quality that could not identified using clinical sleep quality measures.

“In this study our primary aim was to identify if treatment of SDB would improve sleep macro-architecture and our secondary aim was to identify if sleep micro-architecture, which is a more sensitive measure of sleep quality, was improved. We also aimed to identify if any improvement in sleep quality would be associated with improvements in parental reports of sleep, behavior, daytime functioning and quality of life”.

Study methodology is clear. The measures used are reasonable overall but i am not sure of the value of using the ESS-CHAD completed by parents in this population of children. 

The ESS-CHAD has been validated in children as a measure of daytime sleepiness (Janssen KC, Phillipson S, O'Connor J, Johns MW. Sleep Med. 2017 May;33:30-35). It has also been validated in a other studies in children with syndromes such as Prader Willi syndrome. We have added these references to the methods on page 4

“To date this questionnaire has been validated as a reliable measure of self-reported daytime sleepiness in TD adolescents between the ages of 12 and 18 years old [31] and has subsequently been used in children with Prader-Willi syndrome who are also at increased risk of SDB [32].”

There is a lot of data in this study. The layout of the results could be improved perhaps with some clear subheadings so the reader can follow which findings are being described. Perhaps a flow diagram of the treatment or non treatment paths from baseline would help readers to follow the findings. 

Thank you for these comments. We have included more subheadings to assist the reader with interpreting the results and have included a new figure to show the participant flow.

Whilst the authors acknowledge the limitation of small number of participants in their study they still draw a conclusion about treatment effect based on only 9 treated participants, 4 of whom still had severe OSA after treatment so are actually still not technically treated. I think it is wrong to conclude that there is no effect on sleep quality, behaviour and function from just five patients and I am concerned that the paper would be confusing with an unjustified take home message for an inexperienced reader. 

We acknowledge our small sample size and the fact that most children had residual OSA. We have further emphasized the prevalence of persistent OSA in the discussion, and added a sentence to the conclusion stressing the need for larger studies to confirm our findings.

Page 13

“As would be expected clinically the group of children who were treated had more severe SDB than those who were untreated, with almost half (47%) of the untreated children having undergone previous treatment and having PS or Mild OSA at baseline. Although the children who were treated had more severe SDB at baseline, at follow-up overall the OAHI had significantly decreased by around 50% in the treated group from 40.3 ± 46.9 to 17.9 ± 26.9 events/h (p<0.01). In contrast, in the untreated group the OAHI had increased slightly from 4.3 ± 5.7 to 7.3 ± 14.7 events/h, indicating that in some children SDB had worsened (as shown in Figure 2). Our findings of no significant effects on sleep quality, behavior and daytime functioning may have been affected by the fact that the majority of children in both treated (78%) and untreated (73%) groups still having residual OSA. OSA has been demonstrated to affect daytime behavior in children with OSA compared to no OSA [8], however not all studies have shown this [9].”

and in the conclusion on page 16

“In conclusion, our study has identified that treatment of SDB improves severity of the disease as defined by PSG parameters and this was associated with parental re-ports of improved QOL, despite treatment having no demonstrable impacts on sleep quality, behavior or daytime functioning. It must be acknowledged that our sample size was small, and the majority of children in both the treated and untreated groups had residual OSA, necessitating the need for further larger studies to confirm our findings.”

Round 2

Reviewer 2 Report

I think the authors edits have made the paper clearer and have acknowledged the limitations of this study.